# Identifying Shared Strategies and Solutions to the Human–Giant Tortoise Interactions in Santa Cruz, Galapagos: A Nominal Group Technique Application

**Francisco Benitez-Capistros [1],\*, Paulina Couenberg [2], Ainoa Nieto [3,4], Freddy Cabrera [3] and Stephen Blake [3,5,6,7,8]**

[1]  Biomedicine Research Institute (INBIOMED), Central University of Ecuador (UCE), 170201 Quito, Ecuador
[2]  Ecuadorian Ministry of Agriculture and Livestock (MAG), Galapagos District Directorate,
    200350 Puerto Ayora, Ecuador; pcouenberg@mag.gob.ec
[3]  Charles Darwin Research Station (CDRS), 200350 Puerto Ayora, Ecuador;
    ainoa.nieto@fcdarwin.org.ec (A.N.); fredy.cabrera@fcdarwin.org.ec (F.C.); stephen.blake@slu.edu (S.B.)
[4]  Veterinary Faculty, Complutense University of Madrid, 2804 Madrid, Spain
[5]  Department of Biology, Saint Louis University (SU), St. Louis, MO 63110, USA
[6]  Max Planck Institute for Ornithology, 78315 Radolfzell, Germany
[7]  Whitney Harris World Ecology Center, University of Missouri-St. Louis, St. Louis, MO 63121, USA
[8]  WildCare Institute, Saint Louis Zoo, 1 Government Drive, Saint Louis, MO 63110, USA
\*  Correspondence: fjbenitez@uce.edu.ec; Tel.: +593-02-395-27-00 (ext. 5304)

**Abstract:** Conservation conflicts in protected areas are varied and context-specific, but the resulting effects are often similar, leading to important losses for both humans and wildlife. Several methods and approaches have been used to mitigate conservation conflicts, with an increasing emphasis on understanding the human–human dimension of the conflict. In this article, we present a revision of several conservation conflict cases in the management of protected areas, transdisciplinary and participatory approaches to address conservation conflicts, and finalize by illustrating the application of the nominal group technique (NGT) with the case of the human–giant tortoise interactions in Santa Cruz Island, Galapagos. In this article, we demonstrate the use of novel and systematic participatory and deliberative methodology that is able to engage stakeholders in a constructive dialogue to jointly identify and explore options for shared strategies and solutions to conservation conflicts. The results are comparable with other conservation conflicts cases around the world and illustrate the importance of generating legitimatized information that will further help policy and decision-making actions to address conservation conflicts in the management of protected areas.

**Keywords:** tortoises' migration; management of protected areas; rural areas; farmlands; land use change; social–ecological approach; conciliatory approach

---

## 1. Introduction

The conservation of iconic wildlife and landscapes has been a key driver underlying the establishment and expansion of the protected area (PA) system around the world [1]. Today, PAs form the backbone of the biodiversity conservation movement; however, the effectiveness of PAs is highly variable [1–5]. Problems in the establishment of PAs include their fixed spatial boundaries, which are ineffective in managing species with large or changing ranges [6] or PA establishment without integrating socioeconomic and cultural factors of local human communities [7]. Thus, following the changing views of nature and conservation [8], in recent years, PAs have been redefined from more biocentric views on nature (e.g., supplying habitat for wildlife) to more anthropocentric views

of nature, emphasizing the role of PAs in supporting human life, cultural values, well-being, and delivering multiple ecosystem services [1,9]. Accordingly, natural and human systems are treated as one system [10], and the role of local communities has become more dynamic and active in the management and conservation activities of PAs [7,11]. In this regard, an important element is the rapid and ongoing growth of PAs in the world. According to the Convention of Biological Diversity (CBD) and the Aichi Target 11 [12], the global coverage of terrestrial PAs is expected to expand from the current 14,9% [13] to 17% by 2020 [12]. It is very likely, however, that in this context, socioenvironmental conflict will emerge, as conflict is an inherent element in the establishment and management of protected areas [14].

PAs are the stage of conservation conflicts, which are centered around studying the interactions between human and other species (mostly wildlife) and/or ecosystems. However, these interactions encompass many forms and interrelated aspects between human activities and natural systems. Traditionally, conflicts between humans and wildlife have been framed as human–wildlife conflicts (HWC). However, other approaches argue that species are not human antagonists [15] and that conservation conflicts are better understood as human–human conflicts that arise when two or more parties compete for a specific biodiversity-related interest at the expense of the other [16]. Therefore, it is the misalignment of interests, values or actions between individuals and societal groups (collective interest) which originates and defines a conflict situation [17]. Note, however, that although several categories have been proposed to understand the underlying causes of the origin of conflicts, these categories have often failed to capture the complex and context-specific characteristics of conflict [18]. Redpath et al. [19] argued that there would be four main causes underlying conservation conflicts: (1) The different understanding of human–nature relational values, which gives origin to conflicts when the intrinsic and instrumental values of nature are confronted and disputed as two separate solutions, instead of considering that many important concerns about nature could be a shared element to both [20]; (2) the exclusion of stakeholders in conservation planning, which is often related to the presence, imposition or extension of protected areas (PAs), where local communities are removed from their land with no consultation or adequate compensation, thus generating increasing conflicts between the different stakeholders involved (e.g., park managers and farmers) [7,21]; (3) power asymmetries between stakeholders, which are related to particular actors, usually within organizations or institutions (e.g., governments, NGOs), who exert more control over certain resources and use that 'power' to control and take advantage over the control of other resources from other actors [22–24]; (4) historical factors which are on the one hand related to historical perspectives, beliefs and human–nature-related practices that make conservation appear threatening [25] or on the other hand, when new conservation perspectives assume that historical perspectives and human–nature-related practices are threatening [26].

These four main points are at the core of conservation conflicts and allow us to understand that in order to address such conflicts, transdisciplinary approaches that integrate natural, social, and humanity sciences are necessary [19]. Different approaches have been used for addressing conservation conflicts, including policy decision analysis tools [27], game theory [28,29], and participative and deliberative approaches [19,30], which can integrate several methods [31]. In the following subsections, we first revise related cases of conservation conflicts in the management of PAs. Then, we provide an overview on transdisciplinary and participatory approaches to conservation and conservation conflicts; and finally, we present the human–giant tortoise interaction in Santa Cruz Island in Galapagos as a case study. The case study illustrates the use of the nominal group technique (NGT) as a participatory and deliberative method for generating a list of prioritized actions to help with mitigating conservation conflicts.

### 1.1. Conservation Conflicts in the Management of Protected Areas

With growing human populations living alongside different wildlife, PAs, and their particular managements, complexities of conflict tend to vary widely [32]. Dickman [32] argued that social factors, such as perceptions of risk, damage, and responses, need to be studied in order to understand how

these factors influence the conflict. Moreover, according to a revision of HWC by Peterson et al. [33], most conflicts in/adjacent to PAs are defined as human–human conflicts, where the most prevailing wildlife damage to human food resources is caused by iconic mega-herbivores (e.g., elephants, rhinos, hippopotamus), followed by iconic mega-carnivores (e.g., tigers, bears, wolves) and birds [33]. Conversely, the close and frequent contact between people and wildlife also represents a major threat across much of the large iconic species ranges regardless of the existence of PAs [34,35]. Hunting, competition with livestock and land use changes, human encroachment, cultivation, and deforestation are the mayor threats for large herbivores [35]. Indeed, even with well-protected and iconic large herbivores, such as the Asian Elephant (*Elephas maximus*), human–elephant conflicts (HEC) spawn serious conservation and social concerns due to the large-scale crop damages and loss of human lives [36]. Gubbi et al. [37] argued that despite the fact that elephants are largely concentrated in 12 PAs (14,500 km$^2$) in the Karnataka state, the HEC in southern India is challenging due to the country's economic development and the maintenance of habitat connectivity and landscape compatibility for wide-ranging species with slow recruitment rates, such as the elephant, which acts as a key driver for immigration and emigration [36,37].

In the case of large iconic carnivores, such as lions (*Panthera leo*), conflicts tend to be accentuated by retaliatory killings with devastating consequences for the species, such as in West Africa, where the population of lions is estimated to be less than 500 individuals [38]. In this regard, Dickman and Hazzah [39] noted that the intrinsic fear and dread towards specific predator wildlife (e.g., lions, bears, and pumas) often incur in disproportioned retaliations towards the species. The authors pointed out that the degree of fear is often extremely high relative to the number of deaths that might occur [39]. For example, Japan's brown bear (*Ursus arctos*) killed seven people about a century ago and became infamous, contributing to a generalized fear of bears which continues to persist until now [40]. Even when wildlife does not pose a threat to human lives, local people's perceptions tend to shift rapidly as soon as problems with species develop. This is the case of the Andean Bear (*Tremarctos ornatus*), where, although it can be perceived positively by farmers, as soon as problems develop (e.g., cattle disappears), perceptions towards it switch and persecution and poaching follows [41]. Direct wildlife damage to people's property or resources (e.g., crops, cattle) or human life, in addition to leading to retaliatory actions on the part of humans, also involves significant impacts for people and their livelihoods [42].

Economic incentives are often used to increase the tolerance of the presence of wildlife in private lands. A notable example is the payment scheme to Sami pastoralist in Sweden for each certified reproduction of wolverines (*Gulo gulo*) and lynx (*Lynx lynx*). The reproduction of wolverines and lynxes takes place on Sami villages' reindeer grazing lands with surprising success that now exceeds the target of 90 animals per year [43,44]. Another example, is the innovative and successful model in Kenya with the incorporation cultural aspects of the Maasai people into a program where Maasai warriors became "lion guardians" and use radio-collars to "hunt" and monitor lions [45]. The program has not only allowed providing personal benefits to the Maasai warriors but has also increased conservation awareness and appreciation towards a species that was traditionally their enemy [46]. However, as pointed out by Dickman et al. [47], any payment scheme to encourage co-existence needs to be tailored carefully to the individual situation to avoid problems or perverse incentives, and particularly to achieve the desired conservation outcomes and to satisfy the economic and cultural needs of people bearing the costs associated with living with wildlife [47]. Nevertheless, in some cases, economic incentives can also have a negative effect when viewed as a subsidy towards crop and livestock production. Bulte and Rondeau [48] argued that alternatives to economic incentives are best when direct payments to affected communities are based on wildlife abundance rather than on damages [48]. Indeed, one of the major challenges in the management of PAs is the agricultural intensification and land use change around or in the boundaries of PAs. According to Defries et al. [49], identifying opportunities in which ecological functioning of PAs might be maintained with minimum consequences for land use well-being is of primary importance. A noteworthy example involves

the increased habitat connectivity of the giant panda (*Auluropoda melanoleuca*) inside Wolong Nature Reserve, Schuan, China. In this case, the win-win alternative involved the provision of nonagricultural employment opportunities for local communities around Schuan Reserve, simultaneously reducing pressure on the giant panda habitat from fuelwood collection and agriculture while improving local livelihoods [50].

As shown, conflicts in the management of PAs are varied and context-specific, with different factors involved (risk perceptions, damages, attitudes) and several approaches (payment schemes, direct payments, alternative employments) used to understand and encourage co-existence between humans and wildlife. Nevertheless, conflict can escalate when local people feel that their needs or values of wildlife are given priority over their own needs, or when local institutions and people are inadequately empowered to deal with the conflict [51]. Madden [51] explained that in such cases, conflict tends to intensify, becoming not only a conflict between humans (local people vs. authorities) but also between humans about wildlife. Thus, conservation initiatives fail, the economic and social well-being of local people is compromised, local support for conservation decline, and conservation development efforts meant to offset the costs of living near PAs and wildlife may be impaired [51]. In these situations, HWC often involves human–human conflict where different goals, values, wealth, and power relations originate the conflict. Power can be defined as the capacity to mobilize resources. Resources are broadly defined as persons, assets, material or capital, including human, mental, monetary, artefactual, and natural resources [52]. Therefore, power relations in a system, such as in PAs, can be defined as the ability to mobilize people, thereby exerting power 'over' them. Avelino and Rotmans [52] proposed three typologies of power relations, which can produce a state of balance or imbalance of a system: (i) Having power over, (ii) having more or less power, or (iii) having a different power. Explicitly recognizing these three typologies of power relations and their role in conservation management can allow being strategic in the actions to democratize and equalize asymmetrical power relations, such as when conservation conflicts occur [24]. One of the cases that exemplify these power asymmetries occurred in West Kilimanjaro, Tanzania in 2009 when various villagers chased a herd of elephants over a cliff, killing six of them. While elephant killing is often presented as the result of a rising global demand for ivory, Mariki et al. [53] showed that in this case, other factors were involved in the incident. Among these factors were a severe drought in the region, the increasing raided crops and water pipes destroyed by elephants, the growing human and elephant populations in the area, and the conversion of large areas into different types of PAs by conservation NGOs and governmental agencies. Moreover, Mariki et al. [53] argued that conservation in the study area takes place without local communities who have no influence on decision making, which leads to local communities feeling they are being marginalized and disempowered, causing resistance to conservation [53].

However, the relation between power and knowledge might also play an important role in the equalization of power relations in a system. Avelino and Rotmans [52] defined knowledge as 'the mobilization of mental resources (information, concepts, ideas and beliefs) to reach a specific goal', which is by definition an exercise of power. Thus, by constructing and communicating knowledge, we are exercising power. As Avelino and Rotmans [52] explained, this is not only in terms of '*mobilizing mental resources*', but also in terms of influencing how other actors mobilize all the other type of resources (human, artefactual, natural, and monetary). In order to know which resources to mobilize to reach a specific goal, and in order to know how to mobilize these resources, it is necessary to have knowledge about these resources [52]. This last aspect is extremely relevant in the context of a participatory research process and highlights the responsibility and honest attitude that scientist have to embrace while conducting research with social and political implications [54].

*1.2. Transdisciplinary and Participatory Approaches to Conservation and Conservation Conflicts*

The underlying premise of participatory approaches in conservation is that local communities and conservationists have common interests, and conservation scientists can contribute to solutions by providing information to local communities to support them in their decision making [55].

Integrating local knowledge into science-based solutions is of particular relevance to reconcile conservation goals and local and indigenous people with governance [56]. Both multidisciplinary approaches, which involve collaboration between researchers working within different disciplines, and interdisciplinary approaches, in which there is overlap between disciplines in a research agenda, have proven effective conservation tools [57]. New collaborative efforts among scientists, policy makers, and local communities are increasingly regarded as essential to meet conservation objectives [58]. Thus, transdisciplinary and participatory methods have become increasingly widespread within conservation and natural resource management research [19,59–61], with clear benefits in many cases [57,62]. What distinguishes transdisciplinary research and participatory methods is the engagement of stakeholders in the research process. Thus, in the context of transdisciplinarity and participation, the co-production of knowledge is a collaborative process that involves multiple disciplines and stakeholders from several sectors of the society [63] and that fosters knowledge sharing between scientists, decision makers, and the community [64,65]. Additionally, transdisciplinary and participatory approaches to research can encourage environmental policy integration by prompting environmental managers and policy makers to adopt a system approach to decision making (e.g., social–ecological systems) that considers socioeconomic and ecological aspects as equally important [63].

The idea of participation to address conservation conflicts is to find shared solutions between the opposed parties involved in the conflict to encourage coexistence [19]. Nevertheless, one of the most important aspects in participation to reduce conservation conflicts is to build and maintain trust with the involved stakeholders, in particular with landowners and managers. According to Young et al. [66], such trust building requires effort and resources, opportunities for adequate dialogue between stakeholders and willingness to share power in terms of knowledge and policy implementation, in particular when local stakeholders are dependent on and knowledgeable about natural resources [66]. Often, this is the case of agricultural farmers whose activities depend on natural resources and where solutions to potential conflicts with wildlife tend to be more successful and sustainable in the long-term when solutions involve the participation of local stakeholders [25,67]. A good example of participation is the case of the Wood buffalo National Park in Canada, where local stakeholders (hunters, trappers, aboriginals) have been involved in a long-standing cooperative management with the federal government to support decision-making processes and achieve common goals in the conservation management of the National Park [68]. Another example is the participatory process between conservationists and state mangers in the uplands of the United Kingdom to better understand the different perspectives and values about the conflict between hen harrier (*Cyrcus cyaneusi*) conservation and the management of red grouse (*Lagopus lagopus*) for commercial hunting [69]. The role of scientists in participatory processes may also be essential particularly to develop methods to bring stakeholders together and analyze conflicts within a rational framework through multicriteria decision analysis (MCDA) methods [27,31], social–ecological methods and approaches [70,71], and participatory deliberative methods, such as participatory rural appraisal (PRA), focus groups [72,73], and the nominal group technique (NGT), which we present in this paper in the following sections.

### 1.3. Case Study: Human–Giant Tortoise Interactions in Santa Cruz Island, Galapagos Archipelago

In the Galapagos archipelago, the Galapagos National Park (GNP) was established in 1959 in recognition of the unique biodiversity of the islands. Unlike most other PAs around the world, where PAs are surrounded by urban and/or rural areas, in Galapagos, the opposite occurs: PAs completely surround urban and rural areas in the four inhabited islands of Isabela, Santa Cruz, San Cristobal, and Floreana (Figure 1). As a conservation strategy, 97% of the terrestrial surface area of the archipelago (7880 km$^2$) is designated as a national park, positively impacting the protection of species, habitats, and ecosystems [74,75]. Considered a largely pristine "living laboratory", Galapagos was declared a UNESCO World Heritage in 1978. The unique marine and terrestrial wildlife and ecosystems have been the center of conservation research and conservation success for over five decades [76–83]; however, governance and management issues that threaten the biodiversity and sustainability of

the islands remain [84–86], and research priorities such as population growth, climate change, novel pathogens, and invasive species control have been defined as critical to improve the social–ecological fit of conservation strategies in Galapagos [87].

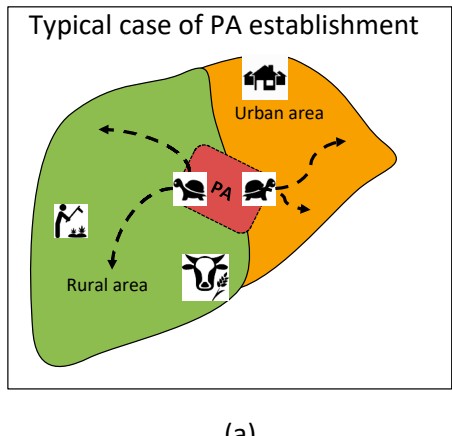 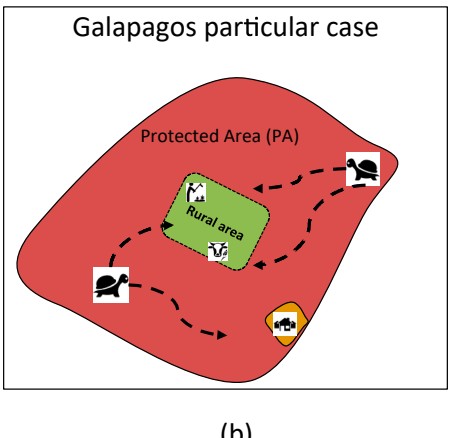

(a)                                                                                      (b)

**Figure 1.** Comparison illustrating the landscape geography of establishment of protected areas (PAs). (**a**) Typical case that establishes PA (**red**) surrounded by urban (**orange**) and rural areas (**green**). (**b**) Galapagos particular case where PA (red) completely surrounds rural (green) and urban areas (orange). The tortoise icon shows different interacting directions (dashed arrows) where wildlife moves outwards from PA to rural and urban (a) and inwards from PA to rural and urban areas (b).

The conservation of Galapagos giant tortoises (*Chelonoidis* spp.) has been largely successful due to a huge reduction in hunting, the eradication of invasive species such as goats and pigs from much of the archipelago, and captive breeding and reintroduction [79,88–90]. Nevertheless, three of the 11 extant species have a close interaction with human activities in two inhabited islands: Isabela and Santa Cruz Islands [91]. On Isabela Island, conservation conflicts have been limited to the occasional killing of giant tortoises due to local beliefs (needs for post-parturition mothers), a cheap but illegal source of food, and retaliations against GNP authorities and conservationists [92]. However, on Santa Cruz Island, the conflict takes a different social–ecological dynamic. The western species of Santa Cruz Island tortoises (*C. porteri*) seasonally migrates from July to December from the national park to upland rural areas where, due to continuously moist conditions, plant productivity remains high throughout the year [93–95] (Figure 2). In common with many migration systems around the world, when protected species leave national parks and enter private lands, the interaction with agricultural and other human activities can lead to conservation conflict [96–98]. Such human–wildlife conflict (HWC) concurrently affects humans (e.g., activities and livelihoods) and wildlife (e.g., safety and survival). However, in both cases, only human responses can originate outcomes to mitigate the interactions and conflict [99]. Thus, it is imperative that an in-depth and joint understanding (e.g., values, goals) about the conflict is achieved between the involved human stakeholders so that solutions can be proposed and implemented [100]. Although disagreements are an inevitable part of human society, the real challenge is to find solutions that minimize their destructive nature and, where possible, create positive outcomes for all actors. While different approaches towards the understanding of socioenvironmental conflicts in and around PAs exist [14], in this paper, we follow Redpath et al. [19] and focus on the human–human dimension of conservation conflicts that addresses the impacts of the direct interactions between human and other species—in this case, the interaction between human and giant tortoises.

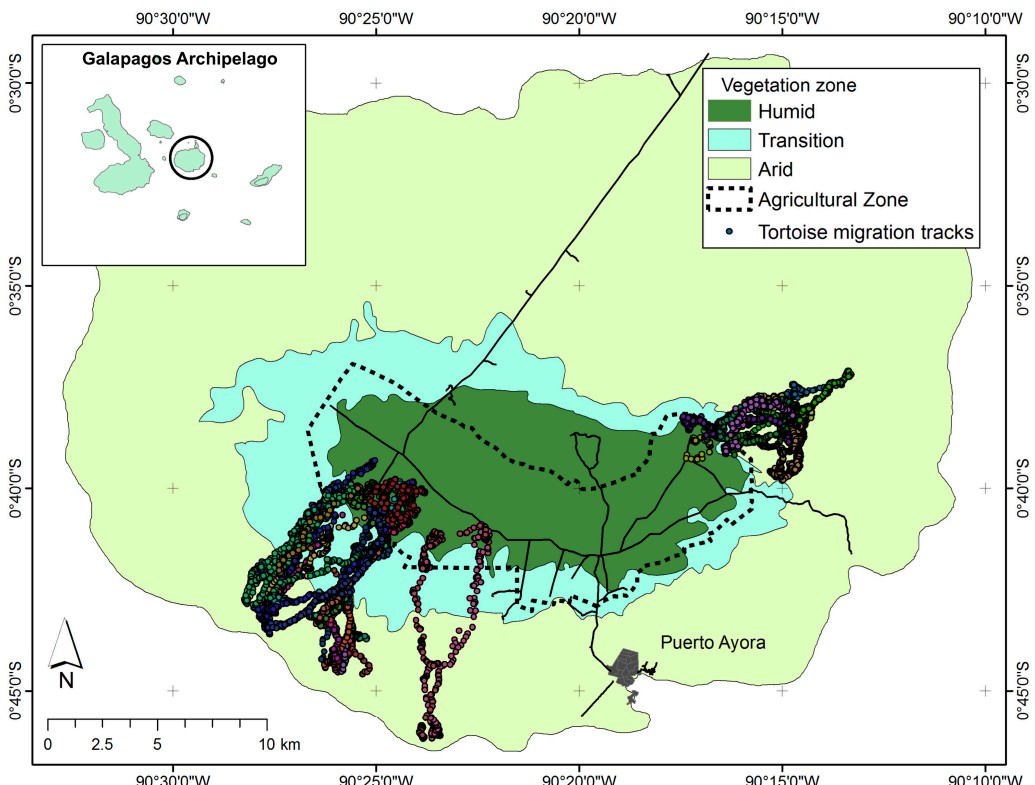

**Figure 2.** Santa Cruz Island, Galapagos. Map illustrating different vegetation zones, the rural–agricultural area, which lies in the center of the island, and the urban area (Puerto Ayora city). A summary of giant tortoise movement tracks in 1 year for the southwestern (*Chelonoidis porteri*) and southeastern species (*Chelonoidis donfaustoi*) are also shown.

In Galapagos, the development of the agricultural areas on Santa Cruz Island has encroached on the migratory routes of *C. porteri* through habitat loss and degradation. Tortoises, in turn, can have negative impacts on the livelihoods of landowners, for example, through damage to crops and fences. A recent study retrieved social (perceptions and attitudes), economic (damage costs to crops and fences), and ecological information (seasonal density of giant tortoises in the rural area) to initially map and begin to understand the conflict [71]. However, in order to start a process leading to more complete understanding of the scale of interactions and integrate related complementary research [93–95,101] into a comprehensible and applicable management framework, a participatory and transdisciplinary process is needed [16,18,19].

The aim of our research was to begin the process of stakeholder engagement in a constructive dialogue to jointly find shared strategies and solutions to the human–giant tortoise interactions in Santa Cruz Island, Galapagos. Using the nominal group technique (NGT), we provide a novel participatory and deliberative methodology to address conservation conflicts in the management of PAs. Consequently, we organized a participatory workshop and used NGT to explore stakeholders' preferences toward: (1) The benefits of giant tortoises for Galapagos; (2) problems facing the continued ecological integrity of giant tortoise migrations on Santa Cruz Island; (3) the problems giant tortoises cause in farms or farming activities; and lastly, (4) a list of prioritized conservation actions/options to the human–giant tortoise conflict in Galapagos.

## 2. Materials and Methods

### 2.1. Participatory Workshop

In the framework of our research project activities with the Galapagos Tortoise Movement Ecology Programme (GTMEP), we held a 1-day participatory workshop in April 2018 in the Santa Rosa parish,

Santa Cruz Island, Galapagos. The aim of the workshop was to involve a group of key actors that are part of the human–giant tortoise interaction in the rural area of Santa Cruz Island. A total of 21 participants attended the workshop (Figure 3), including the organizer and coordinator from the Central University of Ecuador (UCE) and two facilitators from the GTMEP project (also CDRS), and two from the Ecuadorian Ministry of Agriculture and Livestock (MAG). Local actors included 12 local farmers with varied farming activities (e.g., cattle, crop cultivation, tourism, and mixed farming), two delegates of the Galapagos National Park Directorate (GNPD), one delegate of the conservation NGO Galapagos Conservancy (GC), and one additional delegate from the Charles Darwin Research Station (CDRS). The workshop was organized in two phases, each encouraging a constructive dialogue between the participants (Table A1 as shown in Supplementary Materials). The first phase aimed at socializing and presenting different results from our social and ecological research in Galapagos [71,91,93–95,101], the current farming production in Santa Cruz Island and Galapagos, and the current GTMEP, which conducts research on the ecology and conservation of giant tortoises on Santa Cruz Island.

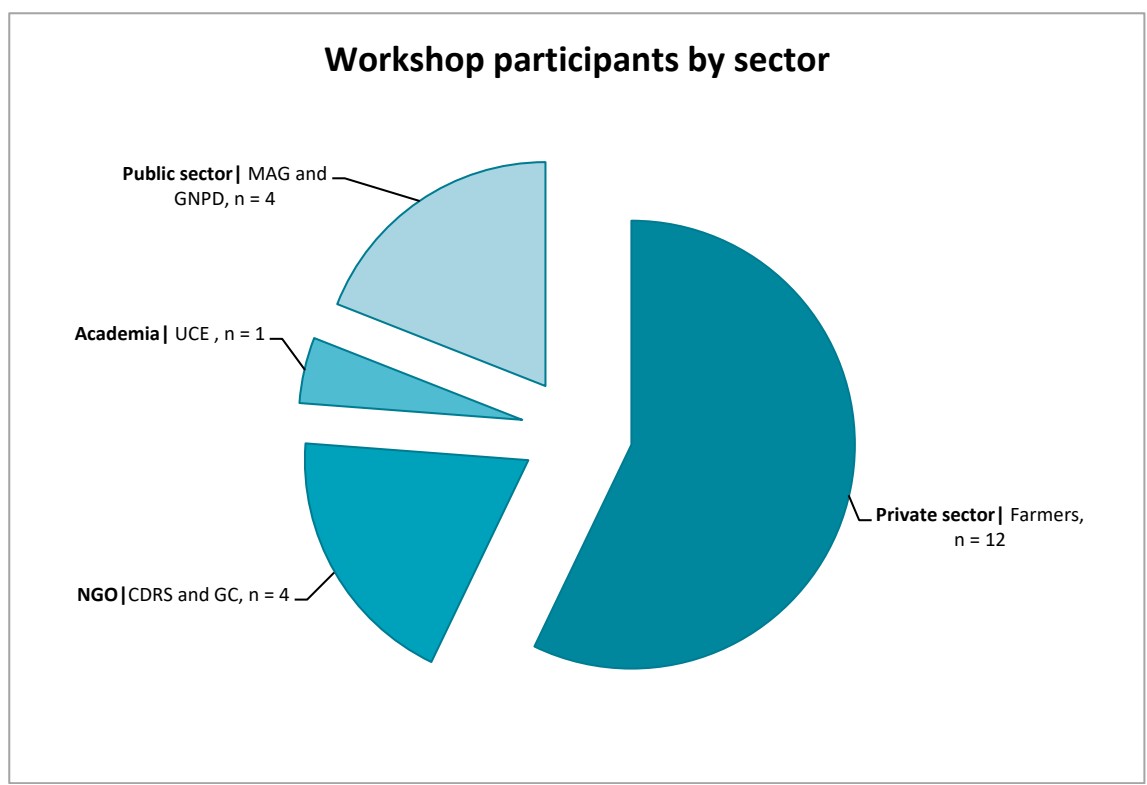

**Figure 3.** Workshop participants by sector.

Informative fact sheets of the different research results/projects were provided to the participants at the end of this phase to provide background knowledge on our research (Factsheets B1–B4), including the ecology of migration and the ecological role of giant tortoises (Factsheet B5). The second phase was more practical and aimed at using the NGT to allow the participants to provide observations, rankings, and opinions about proposed and novel ideas to find co-shared solutions for the human–giant tortoise conflict. F.B-C was the organizer and coordinator of the workshop, while A.N., F.C., and P.C together with a MAG technician were the main facilitators in charge of the technical aspects of the workshop: They ensured that all the NGT steps were followed, maintained consistency during discussions, and helped to interpret suggestions.

## 2.2. The Nominal Group Technique (NGT)

The analysis of conservation decisions requires appropriate use of participatory methodologies that can generate information and knowledge among the different actors (e.g., community, scientists, decision makers) of a social–ecological system (SES). This new information is key to propose solutions, strategies, and actions that will be legitimized by the participation process. NGT is a structured technique based on groups that is used to generate consensus. It consists of four structured steps that allow combining individual and collective reflections to generate a list of priority actions and/or recommendations [102]. In the workshop, NGT was used to generate a list of criteria to promote coexistence between humans and giant tortoises on Santa Cruz Island.

### 2.2.1. NGT Profile of Participants

As shown in Figure 4, the majority of NGT participants were farmers involved in different activities related to the human–giant tortoise interactions in Santa Cruz Island. While this proportional higher number of participants ($n = 12$) was a deliberate choice to ensure the involvement of key local farmers in the process of finding co-shared solutions to the conflict, other relevant actors, such as NGOs officials ($n = 2$) and policy and decision makers, all related to conservation activities ($n = 2$), also participated in the NGT applications. Note that NGT does not aim to map representative opinions among the wider population and is typically conducted with 4 to 20 people [102], which is the same range as that of our NGT application.

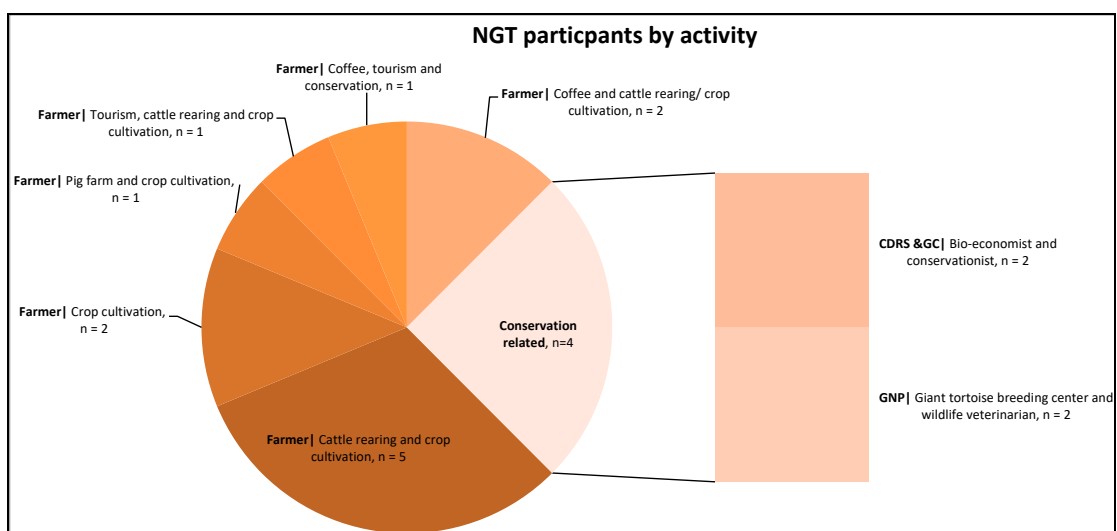

**Figure 4.** Nominal group technique (NGT) participants by activity.

### 2.2.2. NGT Steps

Although typically, NGT involves one or two questions per application [103], in this study, we applied four consecutive NGT applications, since all questions were different but related to Galapagos giant tortoises and the conflict on Santa Cruz Island:

Q1. What are the benefits of the presence of giant tortoises in Galapagos?
Q2. What are the threats to giant tortoise migration on Santa Cruz Island?
Q3. What problems do giant tortoises cause on your land/to your business?
Q4. What solutions, strategies and/or actions would you suggest to allow the migration of giant tortoises to continue and avoid damage to your lands?

Following the outline proposed by Hugé and Mukherjee [102], each NGT application consisted of four steps (Figure 5). In order to facilitate the group dynamic and dialogue and individual generation of

ideas, we divided the workshop participants into three groups, each with one facilitator. Additionally, a goal of the workshop was to share knowledge from previous research results, and thus, we provided a list of statements related to the human–giant tortoise interaction accompanying each NGT question (Table C1–C4 as shown in Supplementary Materials) so that participants could complement, modify or add additional statements, which were later ranked by importance on a 1–5 Likert scale (NGT step 4).

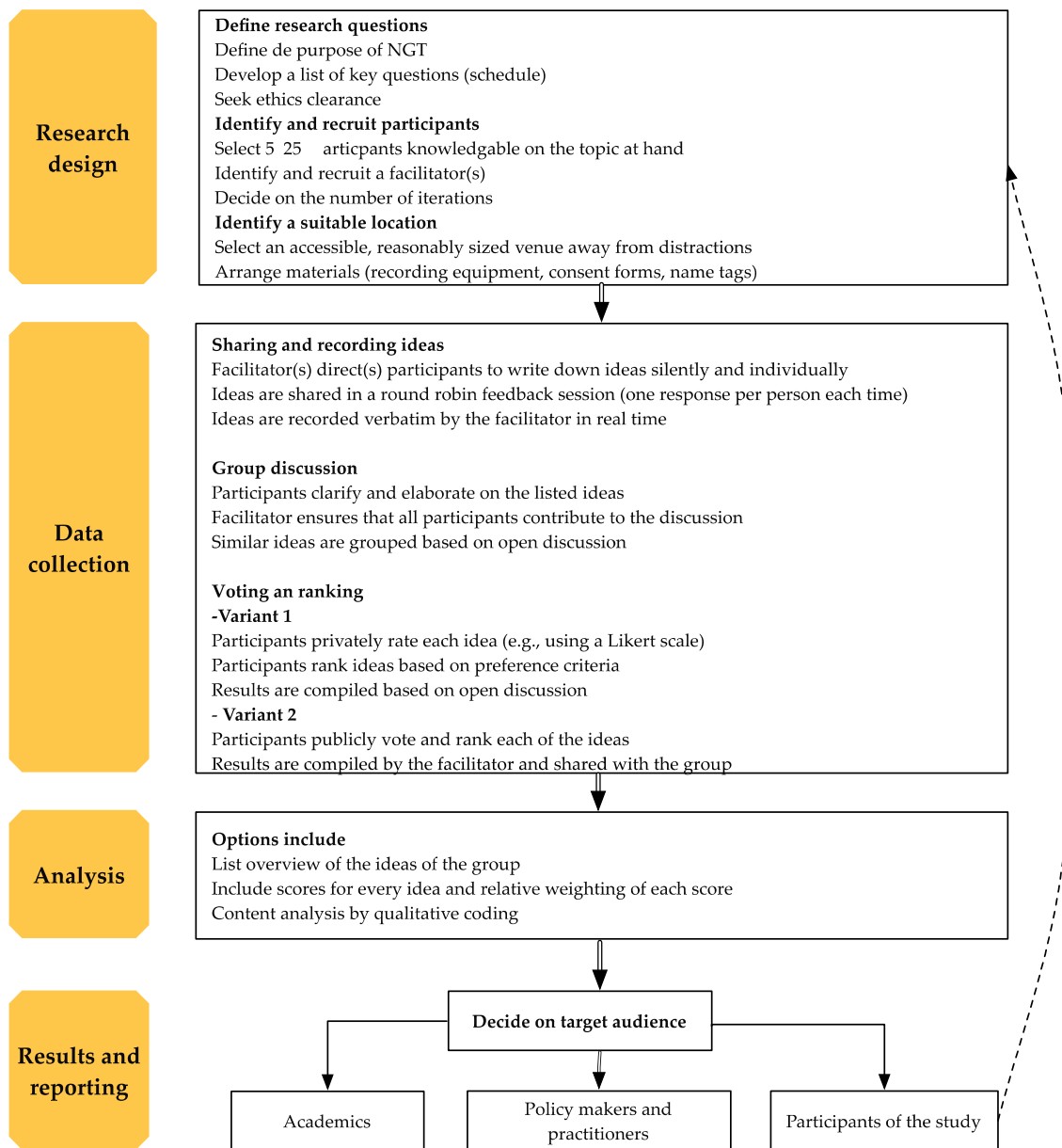

**Figure 5.** Flowchart outlining the steps of a typical nominal group technique (NGT) exercise. Based on and modified from Hugé and Mukherjee [102].

Step 1: Generation of Ideas

In each group table, participants were asked to individually reflect about each question and to write down a range of ideas. At this stage, there was no interaction among participants. This step lasted 45 min.

Step 2: Sharing and Recording of Ideas

In each group/table, ideas were shared by way of a round robin feedback session (one response per person each time) to record each idea consistently. The ideas were all recorded in paper over a chalkboard in each group table, which was later visually accessible to all the participants (Figure 6). This step lasted 45 min.

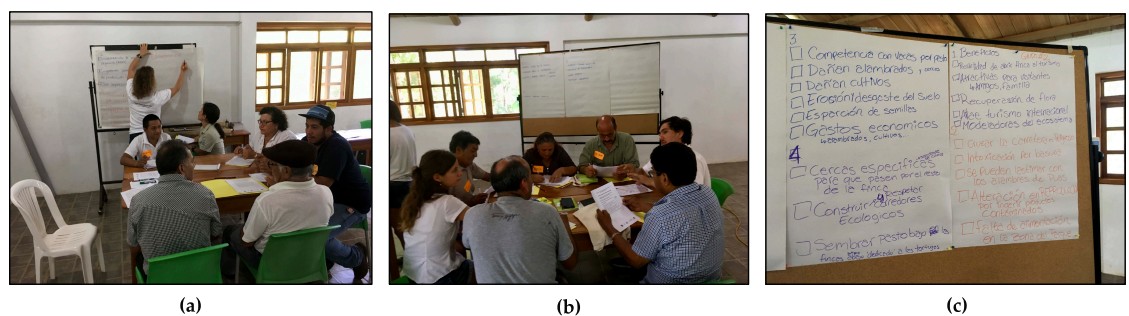

|     (a)     |     (b)     |     (c)     |

**Figure 6.** NGT step 2: Sharing and recording of ideas. (**a**) Facilitator recording group's ideas, (**b**) example of round robin feedback session, (**c**) example of listed ideas in chalkboard.

Step 3: Group Discussion

Participants clarified and elaborated on the ideas proposed in step 2, ensuring that all the ideas were clearly understood. Facilitators made sure that everyone participated in the discussion and helped with grouping the ideas on an open discussion. In this step, there was no ranking or judging of ideas. This step lasted 45 min.

Step 4: Voting and Ranking

In this final step, we asked participants to privately vote and rank each of the listed ideas, including those that were included from our previous research. Then each group/table averaged the individual votes and rankings, which were then shared among all the groups/tables. In this step, we did not restrict votes to top priorities but instead allowed participants to vote and rank each listed idea in accordance with their agreement or disagreement over a 1–5 Likert scale. This step lasted 30 min.

*2.3. Data Analysis*

After the final voting and ranking of the NGT applications, facilitators and the organizer already had an overview of the ideas of the whole group. The group/tables average of votes and rankings were instantly calculated and presented to the participants with a projector. Still, to re-confirm that the participants would keep the results of the NGT application, we later delivered a workshop participation certificate and the listed priority results of the NGT applications. In the Discussion Section, the listed priorities as ranked by the NGT participants are analyzed and compared to research from published conservation research in Galapagos or elsewhere. Top priorities for each question were considered most relevant when the percentage of the total share of votes within the group achieved a super-majority consensus (>75%) [104,105].

# 3. Results

*Results of the Voting and Ranking Exercise*

The results of the NGT are presented as tally sheets, showing the ranked list of priorities for each question: The benefits of giant tortoises (GTs) in Galapagos (Table D1 as shown in Supplementary Materials), the migrations problems of giant tortoises in Santa Cruz Island (Table D2 as shown in Supplementary Materials), the problems giant tortoises cause in the lands/activities (Table D3 as shown

in Supplementary Materials), and the solutions, strategies, and actions to the conflict in place (Table D4 as shown in Supplementary Materials). A tree illustrative summary of the results is shown in Figure 7.

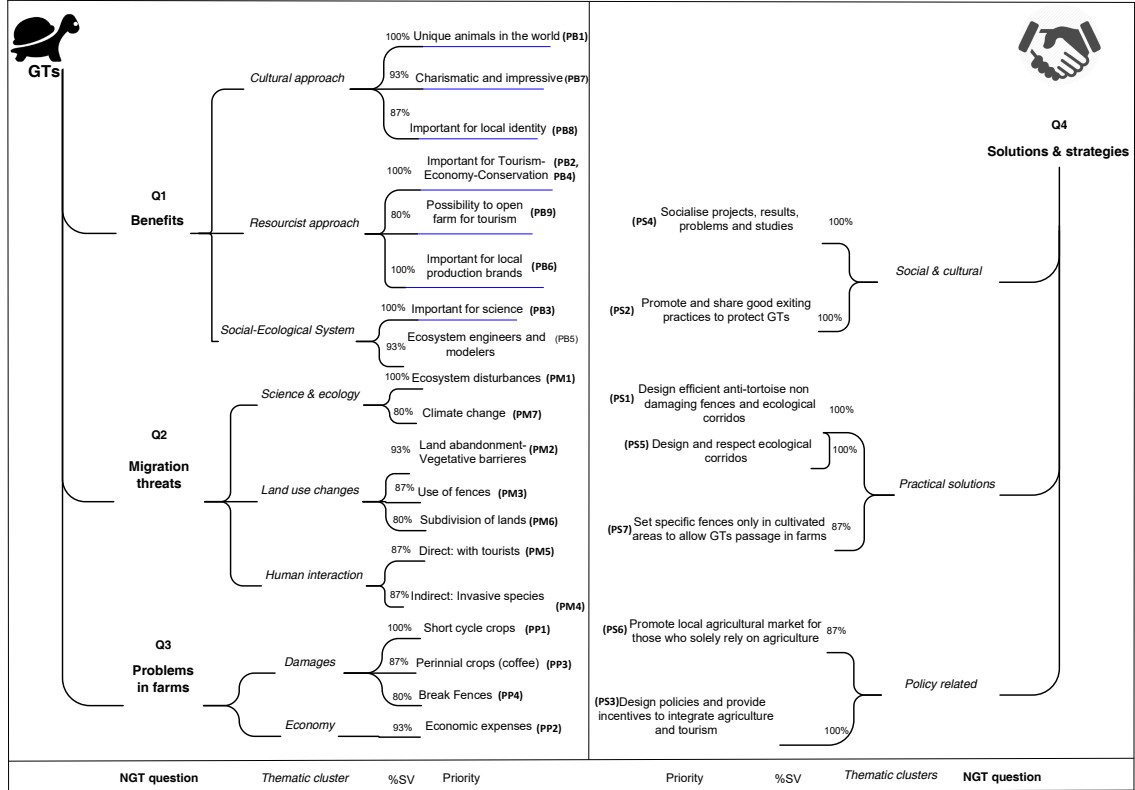

**Figure 7.** NGT tree results with top listed priorities. Left and right panels with NGT questions (Q1, Q2, Q3, Q4), grouped in *thematic clusters*, >75% group's share of votes (%SV), and listed priorities for each NGT application.

## 4. Discussion

The workshop and NGT application explored stakeholders' knowledge and understanding on the benefits of giant tortoises in Galapagos, the potential threats facing tortoise migration in Santa Cruz Island, the problems that giant tortoises cause in their farms or farming activities, and a list of prioritized suggestions for potential solution to the conflict. The structure of the workshop and NGT applications were systematized to allow the flow of information and knowledge sharing between all the involved stakeholders. This allowed us to encourage a more critical and constructive dialogue and discussion about the conflict in place. In the following subsections, we discuss and compare the top listed priorities of each NGT application.

### 4.1. Perceived Benefits of Giant Tortoises for Galapagos

The participants expressed a shared understanding of the benefits of giant tortoises in Galapagos during the NGT-Q1 application. In total, 9 out of 11 priorities were listed as the most important ones, which were grouped into three recognizable clusters. The first benefit cluster (cultural approach) conveys an intrinsic valuation of GTs by the participants. The iconic characteristics of GTs (unique (PB1), charismatic and impressive (PB7), important for local identity (PB8)) show the relevance of these species to mobilize social actions, raise environmental consciousness, and promote public and private interests and economic development in different conservation contexts [106–108]. These results reinforce the few studies that have partially addressed the social and cultural implications of GTs for conservation and the social development of the archipelago [71,82,91].

The second benefit cluster (resourcist approach) combines the perceived importance of GTs for tourism, the economy (PB2), and conservation (PB4). Similar results using a discourse analytical approach determined that indeed, the role of GTs in Galapagos has been shaped by several global conservation governance discourses [91]. Our results also confirm a clear market-driven approach to conservation, with additional highly voted GTs priorities, such as the relevance of GTs to open a farm for tourism (PB9) as well as for local production brands (PB6). These last two benefit priorities (PB9 and PB6) are not surprising. The iconicity of giant tortoises and market-driven approaches to conservation in Galapagos have made the representation of GTs ubiquitous in the archipelago [91]. GTs representations are increasingly used in the branding of local farming products, such as chocolate and coffee, but also with the establishments of economically profitable giant tortoise touristic farms on Santa Cruz Island.

The third and final benefit cluster (social–ecological systems) combines two important listed priorities that describe the importance of GTs for science (PB3) and as the ecosystem engineers and modelers of the archipelago (PB5). Both are intrinsically related and can be interpreted as a consequence of the prolonged efforts in the conservation of the species, which have in turn promoted tourism and conservation awareness in Galapagos [89]. Still, we named this cluster "social–ecological systems" because both priorities (PB3 and PB5) were highly selected during the NGT-Q1 application, highlighting the importance of SES research to integrate social and ecological aspects of systems, to understand their interactions, and to design effective sustainability and biodiversity conservation initiatives [109,110].

### 4.2. The Understanding of Threats Facing Giant Tortoises on Santa Cruz Island

The shared understanding of threats facing GTs migration revealed 6 out of 14 listed priorities in the NGT-Q2 application. We grouped these listed priorities in three clusters. The first cluster (science and ecology), with ecosystem disturbances (PM1) and climate change (PM7), expresses two interrelated variables. Ecosystem disturbances (PM1) can be defined as the events that have triggered discreet to great changes in the structuring of biotics and abiotics elements of an ecosystem [111,112]. Climate change (PM7), however, can be described as a direct consequence of anthropogenic activities (i.e., increased greenhouse gas concentration) that impact physical and biological systems [113], which in turn leads to ecosystem disturbances (PM1) [114]. Both priorities (PM1 and PM7) were discussed during the presentations of the workshop in relation to the spreading of current invasive alien species such as blackberry (*Rubus* spp.), guava (*Psidium guajava*), and passion fruit (*Passiflora edulis*), which have already altered Galapagos ecosystem compositions, since they were brought by humans [115,116]. It is likely, however, that under climatic changing scenarios (PM7), guava and passion fruit will be spread over the national park area in Santa Cruz Island [101], where the main seed dispersers are in fact GTs [94]. With regard to climate change, it is important to realize that climate change has a broad range of structural, psychological, social, and cultural factors that shape people's understanding and perceptions [117–119]. Indeed, different contextual realties shape the general public perceptions about climate change (e.g., alien concept, unknown or risk associated event), and various studies suggest [117,118,120–122] that farmers are well aware of climate change. It seems that farmers have a clearer recollection memory of extreme climatic conditions and other significant events (e.g., prolonged drought, floods, and increased temperature) that have caused disturbances in their productions [123–125]. These events are well known to limit the agricultural productivity in Galapagos [87]. In fact, a drought in 2016 severely impacted farmers in Galapagos, suggesting that many of them are well aware about climate change [126], and that is a possible reason climate change was highly voted by our participants in this NGT-Q2 application.

The second cluster (land use change) encompasses three closely related variables, which have already been demonstrated to be of great concern for the migration of giant tortoises [71]. Land abandonment (PM2) has been identified as one of the most important pressures on Galapagos terrestrial ecosystems [127,128]. While farmers are attracted by better economic opportunities in the prosperous touristic sector, agricultural lands are abandoned, leading to the uncontrolled proliferation

of invasive plants species, such as guava (*Psidium guajava*), cascarilla (*Cinchona pubescens*), and blackberry (*Rubus* spp.) [129–132]. The effects of land abandonment and the proliferation of these invasive plants (PM2) become a three-sided social–ecological problem, affecting: (1) The neighboring farms, (2) the surrounding protected areas of the Galapagos National Park [133], and (3) giant tortoise migratory routes by the formation of vegetative barriers. Whilst the use of herbicides and pesticides to control the propagation of these invasive plants is a common practice, the resulting contamination of the environment is a serious concern [134]. Further research and alternative management actions to deal with this variable will have to be proposed in the near future—in particular, more sustainable and ecologically-friendly practices which are deemed necessary for the fragile island ecosystems. Lastly, the second and third variables (PM3 and PM6) of this land use change cluster refer to the use of various types of fences (e.g., barbed wired, rocks and live fences) (PM3) to stop movement of giant tortoises and other large animals (e.g., cows, pigs) into or out of farmlands. However, fences are also commonly used for subdividing lands (PM6), which can cause direct loss of access to important habitats for tortoises. Greater use of fences could dramatically reduce the habitat area available for tortoises [71].

The third cluster (human interactions) encompasses a variable related to the direct interaction between giant tortoises and tourists (PM5). While the GNPD has set a number of visiting rules to avoid ecosystems and wildlife disturbances (e.g., feeding, touching and keeping distance from wildlife) [135], these often fail to be complied. In particular, this is true for places where less control and regulations exist, such as in the touristic farms of the rural areas. Indeed, in 2017, photographs of a tourist in close contact with giant tortoises was broadly reported over social, local, and national media [136]. Haider [137] argued that social media is a tool that has an important role in shaping the way in which information on environmental behaviors is articulated, shaped, and filled with meaning. The news about the close interaction between wildlife and tourists caused an overall proenvironmental reaction to ameliorate tourists' control. For instance, currently, in one of the four giant tortoise touristic farms, it is required to be accompanied by an authorized GNP guide. It is important to note that close human interaction with wildlife can produce a series of negative effects. For instance, tourism has been proven to significantly alter physiological parameters (endocrine and immune responses) that affect the fitness and survival of Galapagos marine iguanas (*Amblyrhynchus cristatus*) in Santa Cruz Island [138]. Green and Giese [139] mentioned that short- or long-term stressed psychological states of animals can cause increased energy expenditure or reduced breeding success for the wildlife. In particular, the authors argued that for rare or threatened species, even transitory and seemingly minor effects on individuals may be of conservation concern [139].

Likewise, the second variable of this human interaction cluster affecting giant tortoise migration, although indirectly related to humans, refers again to invasive species: Pigs and ants (PM4). Both feral pigs and introduced ants have been well described as human-introduced animals and can have negative effects on the conservation and population status of Galapagos giant tortoises. Feral pigs (*Sus scrofa*) destroy giant tortoises' nests and eat hatchlings [90,140,141]; introduced ants, specifically tropical fire ants (*Selenopsis geminata*), kill the hatchlings of giant tortoises [142,143]. Both variables (PM5 and PM4) were highly voted by our NGT participants and suggest a clear environmental awareness of the secondary human impacts on giant tortoises.

### 4.3. The Understanding of the Problems Giant Tortoises Cause in Farmers' Lands/Activities

The section was grouped in two clusters: Damage and economy. While damages refer to the variables of crops (PP1 and PP3) and fences (PP4), the economy cluster refers to the direct consequences of damage that result in overall/added economic expenses (PP2) for farmers. These highly rated variables during this NGT-Q3 application confirm our former research results [71], where damages and the associated damage costs to short-cycle (PP1) (e.g., corn) and perennial (PP3) crops (e.g., coffee) and fences (PP4) were identified in the area. The estimated damage costs for crops and fences averaged 2.8 USD/m$^2$ and 13 USD/m, respectively. While these costs were only estimated perceptually, this information provides an idea of the associated damage costs that are caused by the presence of giant

tortoises in farmlands [71]. Measurement of real costs of damaged crops and fences is necessary to assess tortoise impacts. Certainly, our approach and results in this paper can already suggest important prioritization actions, which are discussed in the following sections.

*4.4. Proposed Solutions, Strategies, and Actions to the Human–Giant Tortoise Interactions*

The proposed solutions, strategies, and actions to the human–giant tortoise interactions in Santa Cruz Island were grouped into three clusters. Thus, the first cluster (social and cultural) includes the variables: Socialize projects, problems, and results (PS4) and promote and share existing best practices that minimize potential conflicts with giant tortoises (PS2). The first variable (PS4) re-affirms a generalized social demand and interest by the local communities to be involved in or at least aware about conservation-related research [91]. Chase et al. [144] explained that these social demands are common in PAs, where local users often perceive that insufficient opportunities are given to them for participation in conservation and wildlife management [144]. Indeed, during our workshop, participants were happy to be consulted and to learn about current giant tortoise research in Galapagos. The second variable (PS2) shows an intrinsic cultural valorization of the iconic giant tortoises by the local communities who, as shown by Benitez-Capistros et al. [91], consider that the protection of giant tortoises is a moral duty. Most likely, this is the main reason our participants have suggested promoting and sharing best practices to promote conservation of giant tortoises (PS2) during this NGT-Q4 application.

The second cluster (practical solutions) refers to specific hands-on solution to deal with human–giant tortoise interactions. The two proposed solutions in this cluster specified the need to design fences to protect small crop areas (PS7) that do not harm tortoises, in combination with ecological corridors to guarantee giant tortoise migration through the agricultural area (PS1 and PS5). Barbed wire is the predominant type of fencing in Galapagos and the region but is often hazardous for wildlife [145,146]. Alternative fences could include plain high-tensile fencing wire or woven wire instead of barbed wire, which have proven effective in controlling wildlife damages [147]. Note, however, that anything that cuts down on habitat could be considered 'harmful', and thus, we only refer to the use of fences that do not physically harm giant tortoises. Certainly, in addition to finding alternatives to barbed wire fencing, the use of fences only in cultivated areas (PS7) was also prioritized by our participants during this NGT-Q4 application. It is important to note that while alternative fencing could be easier to implement on a personal farmer basis and/or with some policy support, establishing ecological corridors through private lands requires an integrated policy–science–community landscape approach and analysis [148]. While the current telemetry data [93,95] have been key to determine giant tortoises' ecological roles and migratory routes, other landscape and geographic information (e.g., vegetation maps) will be needed in order to determine specific areas where giant tortoises encounter migration obstacles (e.g., fences, vegetative barriers). Furthermore, before initiating the implementation of ecological corridors in the area, it will also be necessary to exactly identify the areas or farms where real verified damages to crops and fences have occurred, as indicated in Section 4.3. Though several methods to establishing ecological corridors exist (e.g., modeling, expert guidance) [149], any approach will require transdisciplinarity to legitimize and sustain the conservation-related action in the long term [148].

The last cluster (policy-related) contains two variables that seek policy support for the agricultural sector in Santa Cruz Island. The first variable, promote local agricultural market for those who solely rely on agriculture (PS6), is rooted in the limited market access for local farmers. It estimated that about 30% of fruits and vegetables and 83% of meat are produced locally to supply food to the residents and tourists in the archipelago [100]. However, commercialization of local produce is mainly through intermediaries who erode farmers' profit, which is already marginal due to high labor and input costs for the control of invasive species [150]. For example, in the case of tomatoes, 78% of the total sold in Galapagos is produced locally, but direct sales by farmers only represent 37% of transactions [151]. Moreover, agricultural production conditions, subsidies to fossil fuels, and existing systems for the sale

of food in the archipelago encourage importation of food from the mainland, with a profit margin of 47% for crops grown in Galapagos (average for 29 products) versus approximately 229% for the same products imported from the mainland [152]. In this regard, the food-supply system model developed by Sampedro et al. [153] predicts that without changes in food policy, agricultural food imported from mainland Ecuador will increase to 95% by 2037. Therefore, policy support to regulate and control food imports should help to avoid market monopolization by retailers and benefit Galapagos farmers with improved market access and better profits.

The second variable specifies the need to design policies and provide incentives to integrate agriculture and tourism (PS3). This variable is connected to the second benefit cluster (resourcist approach) described in Section 4.1 with the variables PB9 and PB6, which indicated farmers' desire to implement giant tortoise touristic farms. Farmers could alternatively consider a community-based wildlife tourism (CBWT) approach, which would ideally offset the higher elevated costs of an individual touristic entrepreneurship. Nevertheless, CBTW farms need a series of additional considerations [154,155], among which, one the most pertinent is that farmers need to understand that CBWT benefits are social (e.g., project for communal school or clinic) rather than financial [71].

### 4.5. Comparison of the NGT Results with Other Cases on Conservation Conflicts in the Management of PAs

While conservation conflicts in the management of PAs are varied and context-specific, the results of our case study highlight some shared elements with other cases outside Galapagos. For instance, the fact that PAs in Galapagos surround urban and rural areas (97%) is already a predictor of conflict, not only because the competition for limited space and resources is higher in small islands [156], but also because land scarcity commonly leads to land use conflicts [157]. In the case of Azores Archipelago, which has a considerable lesser share of its terrestrial surface as designated PAs (>35%), Bragagnolo et al. [158] found that even then, the generalized perception of local stakeholders is that PAs constrain local development [158], something that is also well known to occur in Galapagos [159]. Thus, specificities of a system (e.g., PAs in small islands) can explain certain common aspects of conflicts in PAs. However, our results also suggest that in the case of the human–giant tortoise interaction in Santa Cruz Island, other societal factors are involved, and these are not necessarily negative. For instance, the 'cultural approach' cluster indicates that giant tortoises are perceived as charismatic, impressive, and important for the local identity. This is something different from other cases where dread and fear of wildlife (e.g., bears, wolves, lions) lead to detrimental actions (e.g., killings) towards the species [38,39]. Nevertheless, even if the tame nature of giant tortoises and in general Galapagos wildlife [160] is not associated with fear or dread, the perception of risk from a threat is not the same as being vulnerable to it [32]. Indeed, as problems might develop with wildlife, perceptions can shift rapidly (e.g., conflicts with Andean Bears [41]) and lead to detrimental actions towards wildlife as with the unfortunate killing of elephants in Tanzania [53].

Farmers might not threaten giant tortoises with killing, but the fact that they cause damage to property (fences) and resources (crops) was a highly voted element of the perceived conflict that giant tortoises cause in farmers lands and activities. Giant tortoises can therefore be added to the list of iconic mega-herbivores (e.g., elephants) which, as explained by Peterson et al. [33], are the most prevailing wildlife that cause damage to human food resources in/adjacent to PAs. The challenges involved with giant tortoises are similar to those of the Asian Elephant over in the Karnataka state [37] and the maintenance of habitat connectivity and landscape compatibility will need to be dealt with for the migratory species as well as for the economic development of the local populations. In this regard, while our NGT result offers practical solutions (e.g., design of nonharmful fences to protect small crop areas and establish ecological corridors), landscape and geographical data (e.g., vegetation maps) need to be produced in order to accurately implement them. Interesting approaches from Gubbi [36] and Karanth et al. [161] provide useful applicable examples to understand and map the patterns of human–wildlife conflicts and to provide accurate quantitative information on the conflict.

Similarly, the literature and information available on HWC in/adjacent to PAs shows several successful cases in which compensation schemes have been used to promote co-existence with wildlife [43,44,50]. Nevertheless, our results indicate that while tortoises' damages to crops and fences and the resulting economic expenses are well acknowledged by farmers, desired solutions emerged in the form of policy incentives to promote local agricultural markets and to integrate agriculture and tourism. However, these solutions warrant much more scrutiny, particularly because these can be taken as incentives to convert additional land to agriculture (crops) or to increase the livestock on lands. Bulte and Rondeau [48] have argued that these are indeed common negative effects in the compensation schemes for wildlife damages in agricultural settings. In these situations, compensation for wildlife damages could thus be better articulated by wildlife abundance rather than damages [48]. In any case, there is a strong need of policy support for the implementation of mitigating strategies, which can favor wildlife (e.g., maintaining habitat connectivity) and at the same time satisfy the economic and cultural needs of the people living with wildlife or in/adjacent to PAs [47]. For this to happen and to be sustainable in the long-term, such as with management of the Wood buffalo National Park in Canada [68], the maintenance of lynxes and wolverines in Sweden [43,44], and the increased habitat connectivity of Pandas in Wolong reserve in China [50], a willingness to share power and knowledge among the involved stakeholders is needed. Similarly, in Galapagos, any envisioned alternative to promote co-existence between humans and wildlife will need to address such power asymmetries. The results of this paper and pilot study with the application of the NGT have contributed to building trust and sharing power and knowledge among important groups of stakeholders involved in the conflict.

### 4.6. Limitations of the Nominal Group Technique

In line with Hugé and Mukherjee [102], we agree that the nominal group technique is a rigorous and systematic technique that is able to elicit views and generate opinions and debates from a wide variety of stakeholders. The intrinsic conciliatory approach and depolarizing nature of NGT contributes to addressing conservation conflicts. Nonetheless, it is important to emphasize that the nature of this study is restricted to an exploration and ranking of a small group of stakeholders' priorities about the human–giant tortoise interactions in Santa Cruz Island. In the future, a larger share of stakeholders, including farmers but also decision and policy makers from different governmental bodies, including the Galapagos Government Regional Council (CGREG), directives of the GNDP, Ministries of Agriculture (MAG), Economy (MEE), Environment (MAE), and Tourism (MITUR), will need to be involved. Nevertheless, the obtained results with the use of the NGT have allowed us to gain good insights into human–giant tortoise interactions in Santa Cruz Island. Involving and engaging various stakeholders proved crucial to initially understand their values and perceptions about the conflict but also to define and suggest policies and regulations to create ecologically viable and accepted solutions for conservation–human conflicts. Building on this small but necessary first step, this information will be key for implementing an island-wide, or Galapagos-wide larger effort to correctly manage human–giant tortoise impacts.

## 5. Conclusions

Managing conservation conflicts requires capturing stakeholders' knowledge and values about conservation conflicts. Thus, stakeholders must engage in a constructive discussion to articulate realistic goals and trade-offs to provide legitimized and shared solutions that minimize future conflicts. The participatory workshop with the use of the nominal group technique (NGT) provided a robust method to initially integrate, share, and co-create knowledge and solutions about human–giant tortoise conflict in place. As applied in this research, the systematized NGT applications allowed us to improve communication, thinking, learning, and constructive dialogue among stakeholders. The consensus obtained for each NGT application revealed stakeholders' values, knowledge, and perceptions about giant tortoises, their migrations, problems, the related conflict, and proposed solutions. While certain

priorities might have been voiced in generic terms due to perceptual differences (e.g., climate change), other novel priorities (e.g., human contact with giant tortoises) and recurring ones (e.g., policies to integrate agriculture and tourism, alternative fencing, ecological corridors) provide useful information to initially address the human–giant tortoise interactions in place. The workshop we have reported here was a small but important first step to begin a process of information sharing and solution generation. Future exercises, building from this experience, will allow scientists and policy and decision makers to identify research gaps and provide legitimized policy and decision-making actions to improve the social–ecological fit of conservation strategies to manage the human–giant tortoise conflict in Santa Cruz Island.

**Supplementary Materials:** Supplementary Materials are available online at http://www.mdpi.com/2071-1050/11/10/2937/s1.

**Author Contributions:** Conceptualization, F.B.-C.; Methodology, F.B.-C.; Software, F.B.-C.; Validation, F.B.-C., P.C. and A.N.; Formal Analysis, F.B.-C.; Investigation, F.B.-C., P.C., A.N., F.C., and S.B.; Resources, F.B.C. and S.B.; Data Curation, F.B.-C.; Writing—Original Draft Preparation, F.-B-C; Writing—Review and Editing, F.B.-C., P.C., A.N., F.C., and S.B.; Visualization, F.B.-C.; Supervision, F.B.-C., P.C. and S.B.; Project Administration, F.B.-C.; Funding Acquisition, F.B.-C., S.B.

**Funding:** This research was funded by National Geographic Society grant No. WW-048R-17.

**Acknowledgments:** The authors would like to thank all the participants for their hard work and contributions during the workshop. We all also thank the District Directorate of Ministry of Agriculture and Livestock (MAG) for their logistic support before and during the workshop. In particular, we thank Antonio Rueda and Alicia Maya for their help and technical support in the organization of the workshop. We also thank the anonymous reviewers of this article that have suggested important aspects to improve the quality of this paper. We thank the partners of the Galapagos Tortoise Movement Ecology Programme (GTMEP) (the Galapagos National Park Service (GNPS), Charles Darwin Foundation (CDF), Max Planck Institute for Ornithology (MPI-O), The Institute for Conservation Medicine of the Saint Louis Zoo (ICM), the Galapagos Conservation Trust GCT), The Houston Zoo and The State University of New York School of Environmental Science and Forestry (SUNY-ESF)).

**Conflicts of Interest:** The authors declare no conflict of interest. The funders had no role in the design of the study; in the collection, analyses, or interpretation of data; in the writing of the manuscript, and in the decision to publish the results.

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
