# Peer review of "Identifying Shared Strategies and Solutions to the Human–Giant Tortoise Interactions in Santa Cruz, Galapagos: A Nominal Group Technique Application"

_sustainability, doi:10.3390/su11102937_

Round 1

Reviewer 1 Report

The research is certainly valuable and can be of interest for a broad audience, but unfortunately in the current form the manuscript focuses too much on the case study and the research lacks depth. It is necessary to frame the case study in a theoretical context, such as the conflicts related to the management of natural protected areas, and develop the literature review on this topic, discussing also the results against their theoretical significance.

Currently, the introduction has two parts. The first one establishes the theoretical context (rows 40-60), while the second discusses the case study (rows 62-135). I believe that the first part should be developed more, including a discussion of the conflicts related to the management of protected areas, and identifying better the research goals. Currently, the research goals (rows 137-143) are not making a contribution to the development of the field, and their research depth does not exceed the case study. The second part of the introduction belongs to the methods, as it relates to the case study. It also needs a map indicating the position of Galapagos archipelago to the international audience of "sustainability".

Methods: in addition to adding the description of the case study from the introduction, the methods should also include the current chapter 3.1 of the results. The profile of participants belongs to the methods, as it is not an output of the research (like their opinions).

The discussions also need to be developed. On the one hand, there should be a description of limitations of the research methodology, and possible directions for the future research. On the other hand, after developing the introduction and better stating the theoretical issue addressed by the paper (for example: conflicts in managing the natural protected areas), the results should be discussed against their contribution to enriching this framework. If such a framework is used, a discussion comparing the outcomes with those of similar conflicts identified in the literature would certainly improve the article, and increase its potential audience.

Author Response

Response to Reviewer 1 Comments

We would firstly like to thank the reviewer for points raised, which have improved the quality of the final version of this manuscript. We will try to respond in the clearest manner to all of the suggestion/comments and corrections.

Point 1: The research is certainly valuable and can be of interest for a broad audience, but unfortunately in the current form the manuscript focuses too much on the case study and the research lacks depth. It is necessary to frame the case study in a theoretical context, such as the conflicts related to the management of natural protected areas, and develop the literature review on this topic, discussing also the results against their theoretical significance. 

Response 1:We thank the reviewers for the comments and suggestions made to improve the quality of this paper. We have followed all the suggestions and modified and expanded the introduction into several sections. Thus in the first introductory part we added the explanation and current definition of PAs and reasons why conflicts are likely to emerge [Lines 50-62]. We then continue with and explanation of conservation conflicts and the different approaches that are used to understand them as well as the underlying causes of conflict origin [Lines 64-90]. We finalize this section presenting the structure and logic of the paper [Lines 92-102], which now includes a revision of conservation conflicts in the management of PAs [Lines 104-213], transdisciplinary and participatory approaches to conservation and conservation conflicts [Lines 215-262] and finally the case study with the use of the NGT as novel participatory and deliberative method for conservation conflicts in the management of PAs  [Lines 264-238]. 

Point 2:Currently, the introduction has two parts. The first one establishes the theoretical context (rows 40-60), while the second discusses the case study (rows 62-135). I believe that the first part should be developed more, including a discussion of the conflicts related to the management of protected areas, and identifying better the research goals. Currently, the research goals (rows 137- 143) are not making a contribution to the development of the field, and their research depth does not exceed the case study. The second part of the introduction belongs to the methods, as it relates to the case study. It also needs a map indicating the position of Galapagos archipelago to the international audience of "sustainability".

Response 2:We thank the reviewer for the comments. As mentioned in the first point, we now include an extended introduction that includes a dedicated subsection that revises and discusses conservation conflicts in the management of PAs [Lines 50-213]. Moreover, following the suggestions by the reviewer we also expanded and added a subsection on transdiscilipnary and participatory to conservations and conservation conflicts [Lines 215-262]. In this section, we introduce concepts definitions and examples where the relevance of certain methods have been applied and how with our case study will illustrate and contribute in this field with the application and use o the Nominal Group Technique with the case study in Galapagos [Lines 264-339]. A new map [Lines 313-314] of Santa Cruz Island has been included and shows the different vegetation, rural and urban areas and a summary of the movement tracks for giant tortoises species in Santa Cruz Island. 

Point 3:Methods: in addition to adding the description of the case study from the introduction, the methods should also include the current chapter 3.1 of the results. The profile of participants belongs to the methods, as it is not an output of the research (like their opinions). 

Response 3:We thank the reviewer for the observations. We have now added the section profile of the participants in the methods section [Lines 386-396]. Note however that following the suggestions by the other reviewer we eliminated the table and now provide figures to present the data.   

Point 4. The discussions also need to be developed. On the one hand, there should be a description of limitations of the research methodology, and possible directions for the future research. On the other hand, after developing the introduction and better stating the theoretical issue addressed by the paper (for example: conflicts in managing the natural protected areas), the results should be discussed against their contribution to enriching this framework. If such a framework is used, a discussion comparing the outcomes with those of similar conflicts identified in the literature would certainly improve the article, and increase its potential audience.

Response 4:We thank the reviewer for the observations. Indeed, we now include a sub-subsection of the discussion to specifically indicate the limitations of the methodology and future directions [Lines 737-753]. Note however, that we did present the limitation of the NGT in the former version of the manuscript. However, we now added explanatory lines to indicate future directions [Lines 743-746]. In regards to the discussion and comparison of outcomes with similar conflicts we now provide a full section discussing the similarities and differences with our case study [Lines 679-735].

Reviewer 2 Report

This is a very well written paper identifying shared strategies to the Human-Giant Tortoise interactions in the Galapagos using the Nominal Group Technique. 

I am impressed by the quality of the paper. It is well-organized, well-researched, well-written, and flows very nicely. Making it easy to read and to follow. There may be a few small suggestions which I have attached. 

The only major issues are that there seems to be too much data included. Table 1 can probably be a supplementary material or summarized into a figure or a smaller table. It also seems that Figure 3 isn't completely necessary and could be removed. All of table 3 also seems unnecessary and could be made into a supplementary material since its essentially reduced into Figure 4, which is nicely visualized. 

The only other major issue is with Figure 1. I don't really understand the arrows and what they signify as it is not well-explained. All that is mentioned is "shows different interacting directions". Maybe remove them or have a clearer description in the legend. The use of superscripts in the discussion is also a bit confusing. I am not sure it is needed. I would also recommend adding the Nominal Group Technique in the title since it is a major aspect of this study. 

Author Response

Response to Reviewer 2 Comments

We would firstly like to thank the reviewer for points raised and the encouragement for continuing doing research in this field. We will try to respond in the clearest manner to all of the suggestion/comments and corrections.

Point 1: This is a very well written paper identifying shared strategies to the Human- Giant Tortoise interactions in the Galapagos using the Nominal Group Technique.

I am impressed by the quality of the paper. It is well-organized, well- researched, well-written, and flows very nicely. Making it easy to read and to follow. There may be a few small suggestions which I have attached

Response 1: We thank the reviewers for the comments and support for our work. It is very encouraging to read/hear that we have done a good work with the manuscript.  

Point 2:The only major issues are that there seems to be too much data

included. Table 1 can probably be a supplementary material or summarized into a figure or a smaller table. It also seems that Figure 3 isn't completely necessary and could be removed. All of table 3 also seems unnecessary and could be made into a supplementary material since its essentially reduced into Figure 4, which is nicely visualized.

Response 2:We thank the reviewer for the comments and suggestions. We completely agree with the reviewer. The data is better shown as figure and thus we now changed the table into ‘Figure 3. Workshop participants’ [Lines 358-359] and ‘Figure 4. NGT Participants by activity’ [Lines 394-395]. Regarding Figure 3, we did not eliminate it as we consider that it shows the workflow of the NGT as applied in Santa Cruz, Galapagos. However, we do agree with the reviewer that former Figure 4 (now Figure 7) summarizes the NGT results and that table 3 was too large to be added in the main text. Thus, following the suggestions of the reviewer, we now added the table(s) into supplementary materials (Tables D1-D4) [Lines 473-477]. 

Point 3:The only other major issue is with Figure 1. I don't really understand the arrows and what they signify as it is not well-explained. All that is mentioned is "shows different interacting directions". Maybe remove them or have a clearer description in the legend. The use of superscripts in the discussion is also a bit confusing. I am not sure it is needed. I would also recommend adding the Nominal Group Technique in the title since it is a major aspect of this study.

Response 3:We thanks the reviewer for the observations. However, we have not eliminated Figure 1 because we consider it does visualize the difference of PA establishment in Galapagos vs other settings. However, as suggested by the reviewer we now include a better description of what the dashed lines mean in the figure [Lines 385-386]. Note however that the decision to maintain Figure 1 is also related to the fact that we have followed the advise of the other reviewer and added a revision of conservation conflicts in the management of PAs [Lines 104-213], transdisciplinary and participatory approaches to conservation and conservation conflicts [Lines 215-262] and lastly the case study with the use of the NGT [Lines 264-238]. Moreover, in the discussion we now also added a section discussing the similarities and differences with our case study [Lines 679-735]. Regarding the use of subscript in the discussion, we have followed the reviewer suggestions and removed the subscripts. NGT has also been added to the title of the article [Lines 1-5]. 

Round 2

Reviewer 1 Report

The authors have carried out a substantial review addressing each comment thoroughly and fully. As a result, the readability of the manuscript improved, and in its revised form the article addresses a broader audience. However, these changes were performed in the manuscript, but are not reflected by the abstract so far.

There is one more step to take, meaning adding a little bit to the abstract, especially a discussion on the broader significance of findings, and lessons learnt from the case study that could be applied elsewhere

Author Response

We thank the reviewer for this important observation! We completely agree that the abstract needed to be changed as well. 

Therefore, we have now re-written the abstract to capture all the changes that we made for our manuscripts. Thus the new abstract introduced the the problems of conservation conflicts in PAs, the human-human dimension of the problem and how we have structured the paper: revision of conservation conflict cases in the management of PAs, transdiciplinary and participatory methods and lastly the applicability of our case and obtained results with the use of the Nominal group technique [lines 22-35]